# Test-Bed Performance of an Ice-Coring Drill Used with a Hot Water Drilling System

**An Liu, Rusheng Wang \*, Xiaopeng Fan, Yang Yang, Xingchen Li, Liang Wang and Pavel Talalay**

Polar Research Center, Jilin University, Changchun 130026, China
* Correspondence: wangrs@jlu.edu.cn

**Abstract:** Ice cores from ice shelves contain abundant paleoclimatic information and provide essential information concerned with the prediction of future climatic change and global sea level variations. Efficient retrieval of ice cores is always an engineering challenge in polar ice and marine research. Here, we present design and other information of a new hot-water ice-coring drill used in combination with a hot-water drilling system that provides a rapid and environmentally friendly ice coring system. The coring system shares the surface equipment and hydraulic hose with the hot-water drilling system. Tests with the drill were carried out at an ice drill testing facility, and theoretical estimations were performed to predict the rate of penetration (ROP) and water flow rates. The results indicate the optimal water temperature for ice-coring to be 50 °C, and the most suitable water flow rate to be from 42 L/min to 55 L/min. With those drilling parameters, the maximum ROP is 27.8 m/h and the ice cores are 55–59 mm in diameter.

**Keywords:** drilling engineering; Chinese hot water drilling; ice-coring drill; drilling parameters; coring efficiency

## 1. Introduction

Ice cores of ice sheets/ice shelves contain abundant paleoclimatic information and provide essential information for paleoclimatic reconstruction and prediction of future climatic change and global sea level variation [1]. Efficient retrieval of ice cores is always an engineering challenge in polar ice and marine research. Conventional drilling techniques such as mechanical drilling have been well developed and attempts have been made to apply them to ice-core drilling [2–4]. However, the success of mechanical ice-core drilling is occasionally marginal in terms of (1) ice core quality, (2) drilling fluids, and (3) drilling in warm ice zones [5]. Mechanical ice-core drilling sometimes causes fragmented ice cores, resulting in gaps in the core profiles that preclude continuous stratigraphic analyses [6]. The drilling fluids are often toxic, and thus environmentally unfriendly, and can contaminate the ice cores, precluding quality chemical analysis [7]. When drilling in warm ice zones, mechanical drilling often fails to acquire ice cores [8]. Thermal ice-core drilling is another drilling technology that has been used in polar ice-core drilling [9–11]. For instance, the Russian exploration Hole #5G at the Vostok station, Antarctica at a maximum depth of 2755.3 m was drilled using the thermal drilling technique [12]. The disadvantages of thermal drilling are its inability to drill through ice containing debris, and its low penetration rate, which inhibits its broad applications [13]. Given the various limitations of drilling techniques, mechanical drilling is still always the preferred way to recover a complete ice core, but some other drilling methods can be used when a complete ice core is not required.

Hot water drilling has been generally accepted as the fastest drilling method in polar regions, and abundantly utilized for rapid drilling to the ice bed since the 1970s [14]. For instance, the International Ross Ice Shelf Drilling Project (RISP) used hot water drilling technology to successfully drill through the ice shelf and reached seawater under the ice, generating a borehole of 2743 m depth and on average

91.4 cm in diameter [15]. American researchers have been conducting a drilling project using hot water drilling since 1978. With consistent improvements in equipment and technology, they successfully drilled to a depth of 2200 m in 2003 [16]. The United States initiated a project called "Antarctic Muon and Neutrino Detector Array" (AMANDA) in 1991, which aimed to drill a hole of 2400 m in depth in the season of 1997–1998, using hot water drilling [17]. Due to the fact that hot water drilling is a full face drilling technique, no ice cores have been acquired in the above mentioned projects. To get ice cores, some modifications have had to be made to meet the requirement: an additional core barrel has been added to the original structure of hot water drill rigs. This drilling technology is called hot water ice-core drilling, and it has unique superiorities: ice cores can be recovered at any desired depth without coring through the total ice layer, with minimum impact on the local environment [18].

Hot water ice-core drilling can be a productive supplement for mechanical drills. It has been used many times to recover short sections of ice for quite specific purposes. In the 1993/94 season, 18 ice cores up to 2 m long and 70 mm in diameter were obtained with a hot water coring drill at zones between Ice Stream B and the Unicorn [19]. Researchers from the California Institute of Technology have used hot water ice-core drilling with an annulus of hot water jets to melt out a cylindrical ice core, and successfully retrieved several ice cores at Siple Dome during the austral summer of 1997/1998 [20,21].

This paper introduces a kind of hot water ice-coring drill used in combination with the Chinese Hot Water Drilling System on the Amery Ice Shelf, which aims to meet the need of Chinese hot water drilling to get ice cores at certain depths on the ice shelf. The research discusses the influence of different drilling parameters on coring efficiency and obtained the optimal coring parameters, and can help hot water drilling to obtain quality ice cores in the future field exploration.

## 2. General View of Chinese Hot Water Drilling Engineering on Amery Ice Shelf

This project was initiated in 2015 by the Polar Research Center of China, allied with other organizations Jilin University, Hanzhou Electronic Science, and Technology University, and aims to use hot water drilling to rapidly drill through the 1500 m thick ice shelf and then deploy oceanographic instruments through the drill holes, establishing an monitoring platform for interplay of the ice sheet–ice shelf–ocean [22]. Figure 1 shows the overall structure diagram of the hot water drilling system, which consists of six main parts, including the drilling system, winch system, water heating system, water circulation system, control system, and auxiliary system. In 2017, the laboratory tests of the system were successfully performed in the Key Lab of National Research Ministry (Figure 2), located in Jilin University, which was a significant step, indicating that the Chinese Hot Water Drilling System possessed the ability to drill through the ice shelf [23].

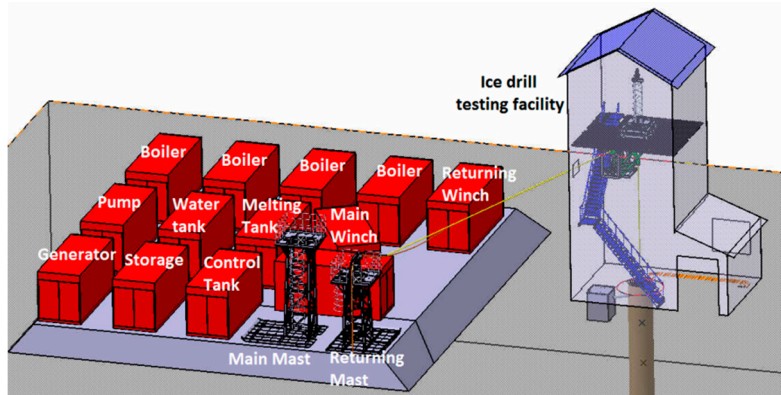

**Figure 1.** Schematic diagram of the overall hot water drilling system.

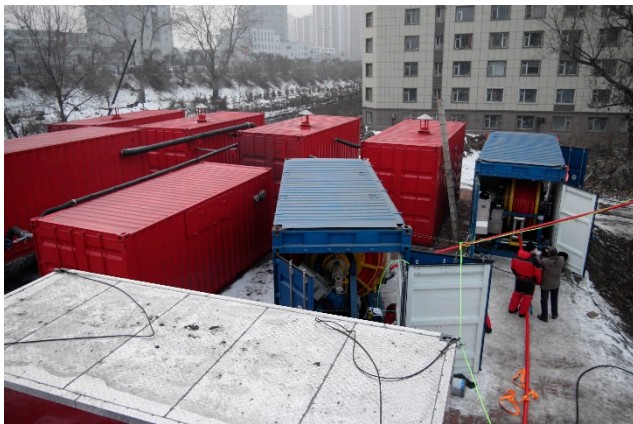

**Figure 2.** Chinese hot water drilling engineering.

The hot water ice-coring drill mentioned in this paper will be used in combination with the Chinese Hot Water Drilling System, which shares the same surface equipment and hydraulic hose for hot water drilling. When ice cores at certain depth are desired, a borehole of that depth should be first accomplished by the hot water drill, which is then withdrawn and replaced with a hot water ice-coring drill. The hot water ice-coring drill will substitute for the hot water drill to go on drilling and coring. Figure 3 shows how the hot water drill and the coring drill cooperate with each other. The following section gives the details of the hot water ice-coring drill.

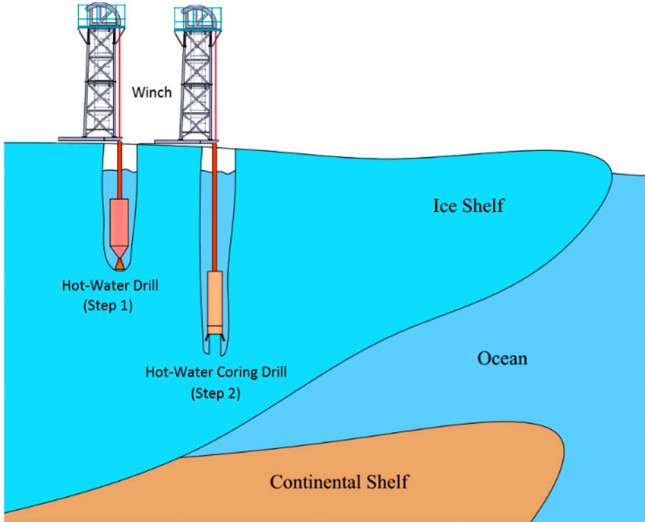

**Figure 3.** Working schematic of hot water ice-coring drilling.

## 3. Design of the Hot Water Ice-Coring Drill

Based on the concept discussed above, the drill was designed, a core barrel of 3 m in length and 96 mm in inner diameter was equipped, and a cutting head was fitted at the end of the core barrel that can spray out hot water. The cutting head consisted of 36 nozzles, each one 1 mm in diameter on the annulus with an outer diameter of 130 mm. Along the outside of the core barrel there were four insulating tubes of 4 mm in inner diameter for hot water supply. The objective for such a design was to prevent the ice core from melting away due to the hot water injection. Four core dogs were bolted in the inner wall of the melting head to capture the ice core when the drill is lifted from the bottom of the borehole. The hot water from the hydraulic hose goes into the connector and then runs through the back drill head and conical bar. Later, the hot water goes through the four water tubes and finally

reach the cutting head, spraying and melting out a cylindrical ice core. A detailed sketch of the coring drill is shown in Figure 4.

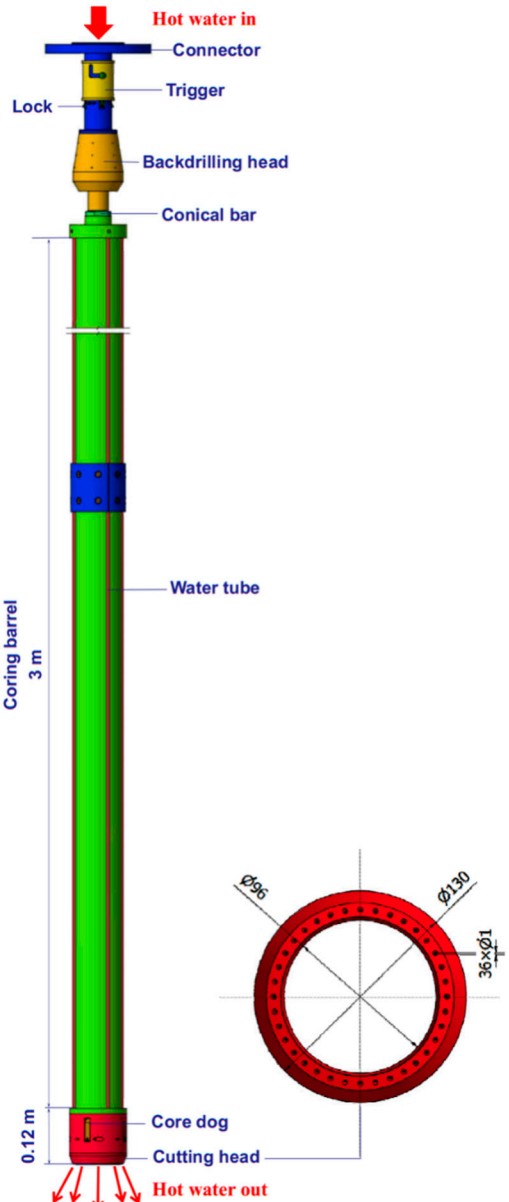

**Figure 4.** Overview of the hot water ice-coring drill.

Sometimes there will be a borehole closure because due to freezing of the remaining water in the drill hole, which complicates lifting the drill, so a back drill head with many nozzles to remove borehole closures was designed. The back drill head possesses a mechanism that can automatically redirect the hot water to flow upwards in case the borehole is blocked due to refreezing during back drilling. A lock was used to fix the position of the conical bar, which was used to connect the core barrel. When the drill goes down, the trigger is opened by hand to release the lock. Due to gravity, the conical bar connected to the core barrel will drop to the coring position. At the coring position, the hot water from the inlet goes through the conical bar to a hermetic inner part, subsequently back to the conical bar, and then is divided into four streams through four water tubes to the melting head, spraying out and melting ice (Figure 5a). When the coring barrel is full of ice core, the core barrel and conical bar cannot move down anymore, but the back drill head will still go down until it reaches the

back drilling position. At this position, the lock will lock the conical bar again. The water then runs through the conical bar to an open inner part of the back drill head, and then reaches the back drill head, spraying out and upwards through the nozzles to melt upper ice (Figure 5b). After this step, the drill can be easily pulled back and the coring catchers will capture the core, break it free, and prevent it from creeping down until the drill reaches the surface.

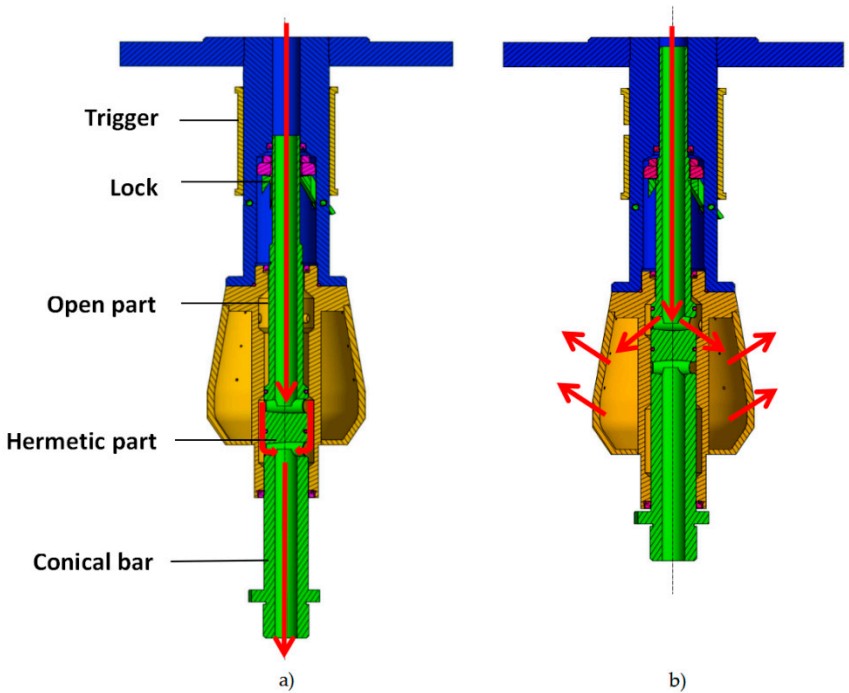

**Figure 5.** The two drilling positions: (**a**) the coring position; (**b**) the back drilling position.

## 4. Theoretical Estimations of the Hot Water Ice-Coring Drilling Parameters

Generally, there is an energy balance at the bottom of a borehole during drilling [24]. Relationships among drilling parameters such as water flow rate, water temperature, and penetration rate during drilling are established from the energy balance. Hot water sprays out from the nozzles and melts ice at the bottom. The ice becomes water, which mixes with the hot water, and all the water then becomes the same temperature. In order to estimate these parameters, some assumptions have been made: no change in physical properties (density) of water and ice occurs over the range of temperature and pressure [25].

The energy during drilling can be divided into three parts:

(1) The energy input from hot water entering through hose, $E$:

$$E = Qt\rho_w c_w T \tag{1}$$

where $Q$ is the flow rate of water, L/min; $t$ is the time, s; $\rho_w$ is density of water, 982 kg/m$^3$ (chosen at ~60 °C); $c_w$ is specific heat of water, 4170 J/kg °C; $T$ is initial water temperature, which is dependent on length and thermal conductivity of hose, °C;

(2) The energy used to melt ice, $E_1$:

$$E_1 = \pi \left( \frac{D^2 - d^2}{4} \right) z \rho_i (c_f - c_i T_i) \tag{2}$$

where $D$ is the diameter of borehole, m; $d$ is the diameter of ice core, m; $z$ is the depth, m; $\rho_i$ is the density of ice, 920 kg/m$^3$; $c_f$ is the heat of fusion of ice, $3.35 \times 10^5$ J/kg; $c_i$ is specific heat of ice, 2100 J/kg °C; and $T_i$ is the temperature of initial ice, assumed to be −20 °C in the ice drill testing facility;

(3) The energy in the final mixture, $E_2$:

$$E_2 = (Qt + \pi az(\frac{D^2 - d^2}{4}))\rho_w c_w T_w \tag{3}$$

where $a$ is ratio of specific volume of water to that of ice (0.92, dimensionless); $T_w$ is final temperature mixture of melted water, assumed to be 20 °C (there is a temperature sensor near the pump; the sensor showed the temperature of melted water to be pumped out was around 20 °C during testing);

According to the energy balance equation:

$$E = E_1 + E_2 \tag{4}$$

we have,

$$Qt\rho_w c_w T = (Qt + \pi az(\frac{D^2 - d^2}{4}))\rho_w c_w T_w + \pi(\frac{D^2 - d^2}{4})z\rho_i(c_f - c_i T_i) \tag{5}$$

and

$$v = \frac{z}{t} \tag{6}$$

$v$—average rate of penetration, m/h;
After dividing Equation (5) by $t$ and solving for $v$, we have

$$v = \frac{4}{\pi(D^2 - d^2)} \frac{Q\rho_w c_w(T - T_w)}{a\rho_w c_w T_w + \rho_i(c_f - c_i T_i)} \tag{7}$$

Substituting in all the values in Equation (7), we have

$$v = 0.00078\frac{Q(T - 20)}{(D^2 - d^2)} \tag{8}$$

## 5. Testing Procedure

For research on the influence on coring efficiency of various drilling parameters (flow rate, water temperature), tests with varying drilling parameters were performed, which aimed to evaluate the efficiency, and furthermore, obtain optimal drilling parameters for achieving a high ROP (rate of penetration), as well as the biggest core diameter.

Figure 6 shows the schematic diagram of the testing. The drill was suspended over the ice well using the wire rope of the rotary drilling platform. The wire rope passed through the pulley and connected with the winch, which was manipulated by a control box by changing the frequency to control the rotation winch, and so adjusting the lifting and lowing speed of the drill.

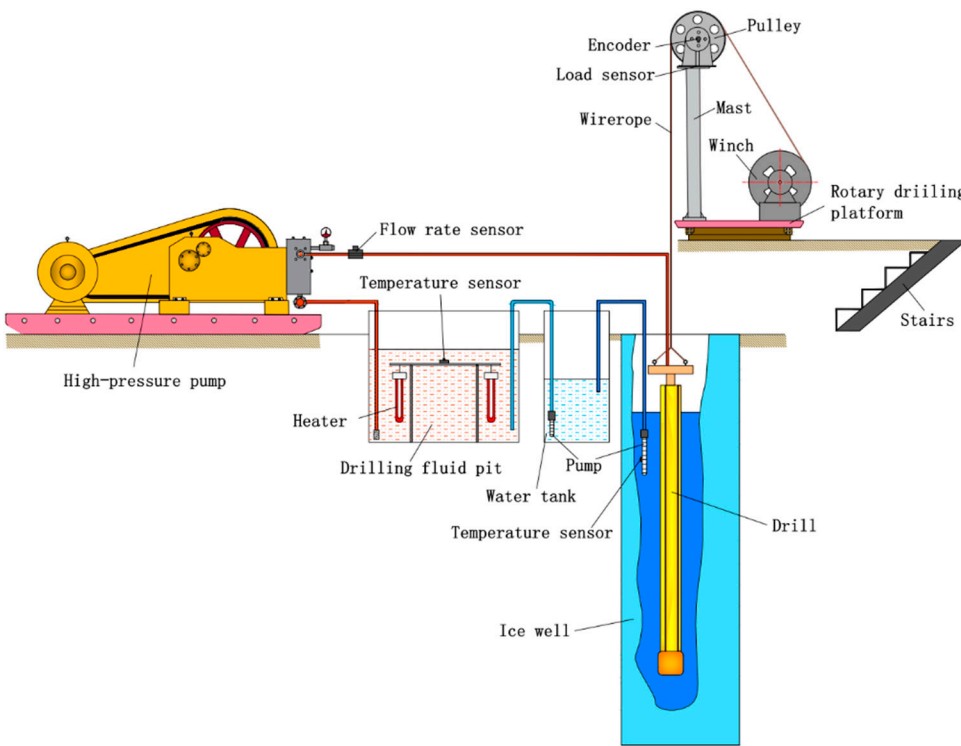

**Figure 6.** Schematic of testing hot water ice-coring drill.

　　Water in the drilling pit was heated by a heater and then pumped through a hose by a high-pressure pump into the drill, spraying out for penetration. When the drilling ended, the submersible pump in the ice well collected water into the water tank. Later, the water will be pumped into the drilling pit and heated again for the next drilling. There were several kinds of sensors, including a LWGB-DN25 flow meter with a measurement flow rate range of 2–160 L/min under a maximum flow pressure of 6.3 MPa at the outlet of the high-pressure pump, a MIK-LCLY load sensor under the pulley with a measurement up to 200 kg, and a HSTL-103 temperature sensor (range from 0 °C to 100 °C) at the top end of the drill. The load sensor was used to monitor if the drill touched the ice. When the drill descends too rapidly and touches the ice, there is a load change, and the downwards speed can be adjusted according to the feedback by controlling the winch to maintain a balanced state. The temperature sensor in the drilling fluid pit was used to measure the temperature of the heated water. Another temperature sensor near the pump was used to measure the temperature of melted water being pumped out. Data including rate of penetration and drilling depth were obtained by the encoder installed on one side of the pulley. We used a plunger pump with power of 18 kW, flow rate of 150 L/min. The maximum pressure was 5 MPa. The total weight of the pump was 1170 kg. Four electric heaters were used to heat hot water, each weighing 2 kg with a wire of 12 m (Figure 7).

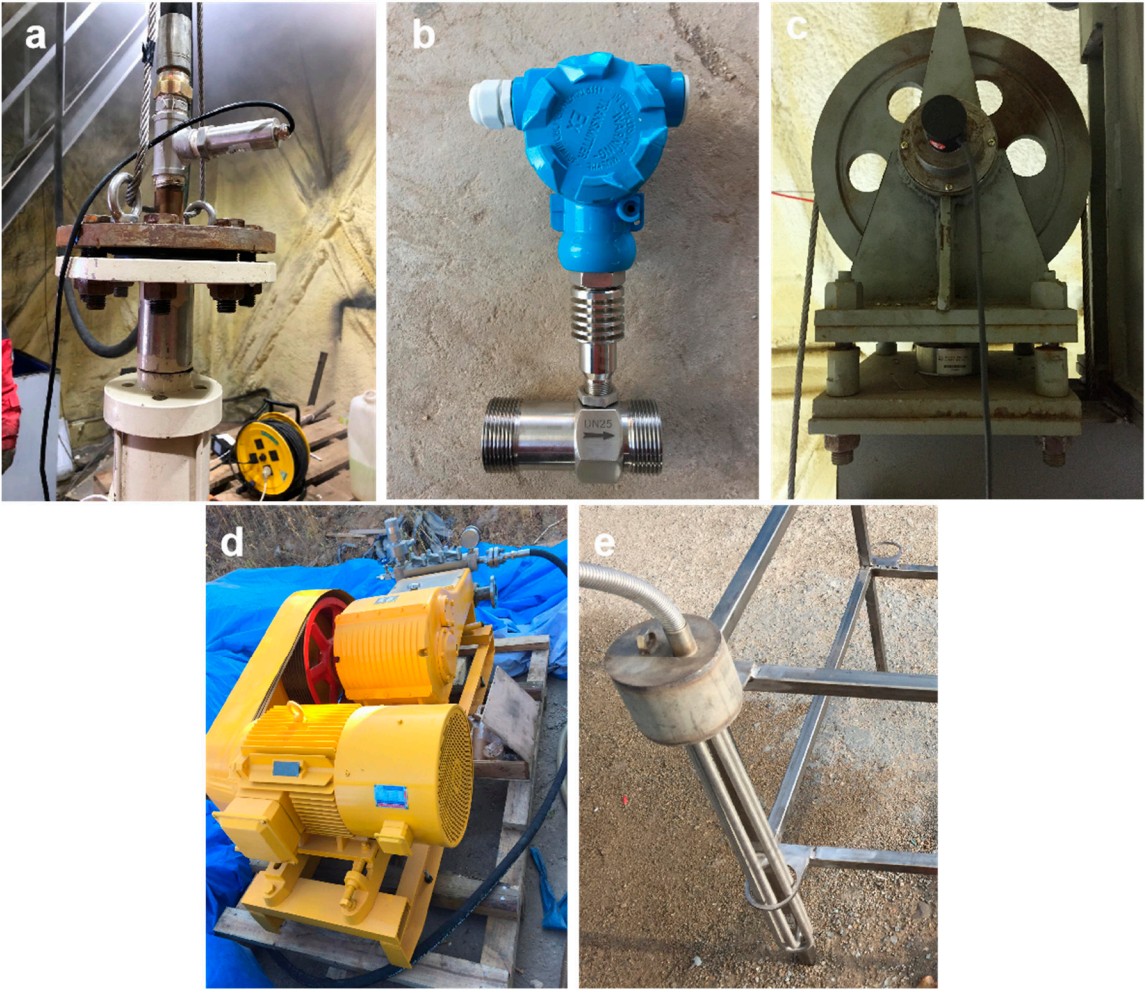

**Figure 7.** Different kinds of devices: (**a**) temperature sensor; (**b**) flow sensor; (**c**) encoder and load sensor; (**d**) pump; (**e**) heater.

A total of 20 drilling runs were performed in the ice drill testing facility, and Figure 8 shows how we drilled.

The size of the ice cores drilled and diameter of boreholes recovered were measured (Figure 9).

Three tests (run No. 1, 2, 3) were performed at different temperatures (50/60/70 °C) of hot water with nearly constant flow rate (55 L/min). Results from these tests indicated 50 °C was the optimal temperature to get ice cores of good quality with larger diameters. During the five subsequent runs (run No. 4, 5, 6, 7, 8), the temperature of 50 °C was kept, but drilling was performed at different flow rates. Five tests (run No. 9, 10, 11, 12, 13) at 60 °C of hot water and five tests (run No. 14, 15, 16, 17, 18) at 70 °C of hot water were also performed with different flow rates (33–52 L/min). All tests were performed when the ice temperature was −20 ± 2 °C.

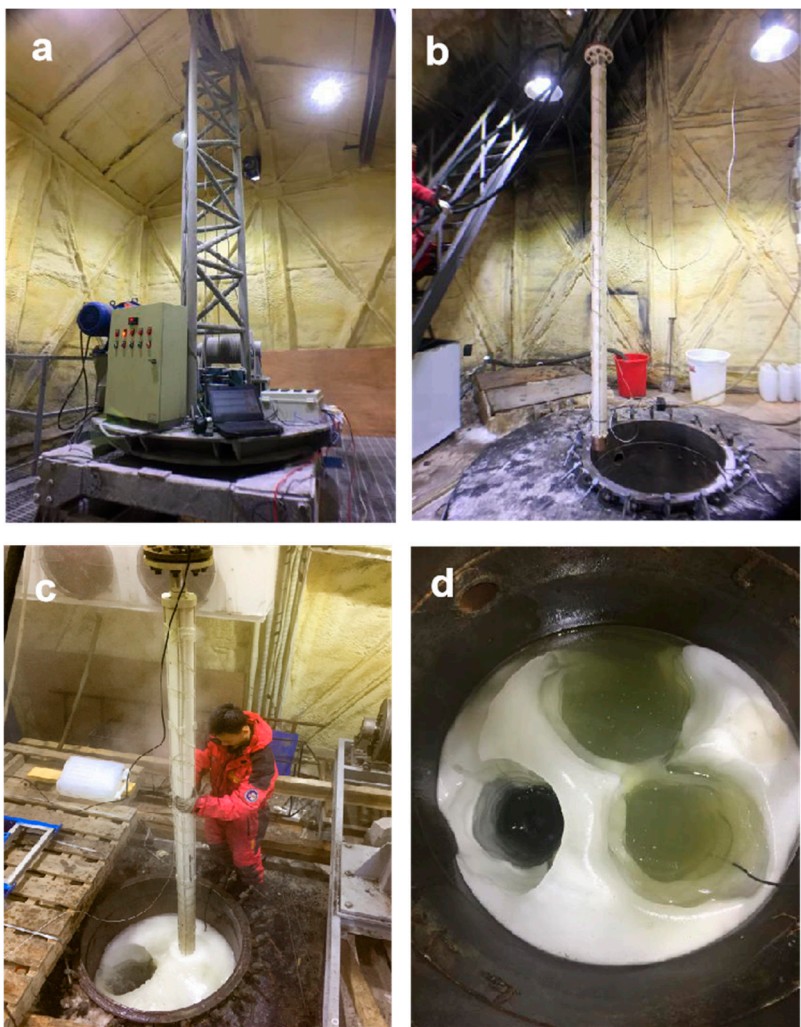

**Figure 8.** Pictures during testing of the hot water ice-coring drill: (**a**) control and data collecting system; (**b**) the drill suspended on the top of the ice well; (**c**) drilling; (**d**) shape of borehole after drilling.

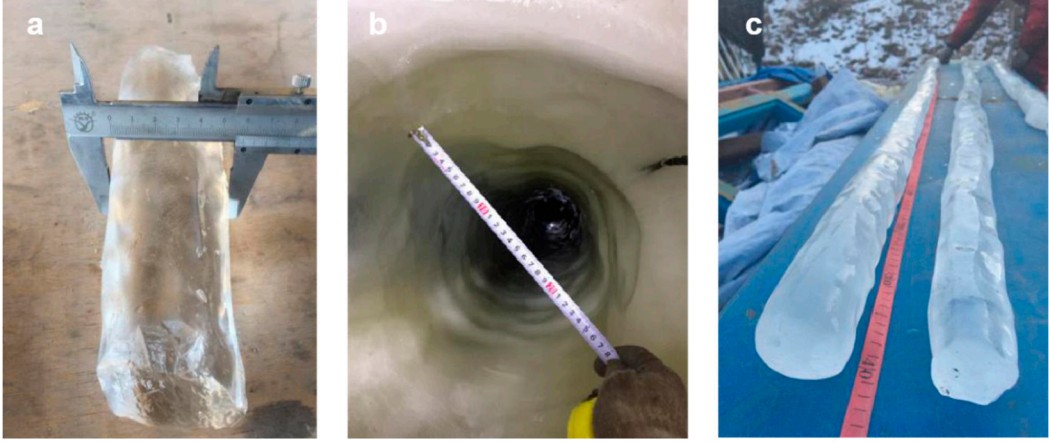

**Figure 9.** Measurements for: (**a**) diameter of ice core; (**b**) diameter of borehole; (**c**) length of ice core.

## 6. Test Results and Analysis

Table 1 shows the test results for the 18 drilling runs.

**Table 1.** Main results of the hot water ice-coring drill tests.

| Avg Water Temperature (°C) | Avg Water Flow Rate (L/min) | Mean Borehole Diameter (mm) | Max Core Diameter (mm) | Avg Core Diameter (mm) | Length of Core (m) | Avg ROP * (m/h) | Run No. |
|---|---|---|---|---|---|---|---|
| 72 | 53 | 271 | 62 | 49 | 2.8 | 27.7 | 1 |
| 63 | 58 | 269 | 60 | 51 | 2.42 | 27.3 | 2 |
| 50 | 55 | 257 | 63 | 58 | 2.58 | 23.3 | 3 |
| 50 | 54 | 253 | 56 | 47 | 2.55 | 21.8 | 4 |
| 50 | 45 | 241 | 64 | 59 | 1.31 | 18.6 | 5 |
| 50 | 34 | 223 | 43 | 40 | 2.48 | 15.6 | 6 |
| 50 | 46 | 227 | 71 | 57 | 2.57 | 20.4 | 7 |
| 51 | 43 | 229 | 60 | 56 | 1.97 | 17.9 | 8 |
| 60 | 50 | 263 | 57 | 51 | 2.23 | 23.1 | 9 |
| 60 | 43 | 253 | 61 | 54 | 1.98 | 20.1 | 10 |
| 60 | 33 | 245 | 52 | 47 | 2.57 | 17.5 | 11 |
| 60 | 45 | 257 | 66 | 55 | 1.75 | 22.7 | 12 |
| 60 | 44 | 259 | 70 | 55 | 1.65 | 20.5 | 13 |
| 67 | 52 | 273 | 65 | 45 | 1.24 | 25.9 | 14 |
| 69 | 41 | 269 | 44 | 43 | 2.61 | 23.2 | 15 |
| 69 | 48 | 278 | 54 | 46 | 2.34 | 24.3 | 16 |
| 69 | 39 | 275 | 57 | 45 | 1.64 | 21.6 | 17 |
| 68 | 34 | 271 | 59 | 44 | 2.42 | 18.6 | 18 |

* Note: ROP is the rate of penetration.

Using the data in Table 1, the relationship between the average diameters of ice core recovered and the water flow rate at three water temperatures is shown in Figure 10. Diameters of the ice cores obtained at the water temperature of 70 °C were generally smaller (maximum 49 mm) than the other two cases. For ice-coring drill, water of 70 °C allowed quicker drilling, but it also melted the ice core faster, resulting in ice cores smaller in radius than the other two temperatures. It was also found that water temperatures of 50 °C and 60 °C could both collect good ice cores (with biggest diameter of 50–59 mm) when the flow rate was 42–50 L/min, and it was obvious that 50 °C retrieved the best ice cores. It was observed that at each hot water temperature, the diameter of the ice core obtained increased as the water flow rate increased, but after a certain flow rate, it began to decrease as the water flow rate continued to increase.

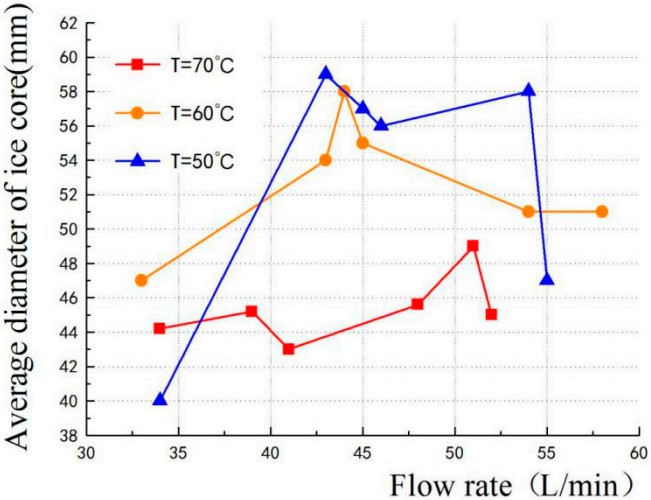

**Figure 10.** Average diameter of ice core versus water flow rate at different temperatures.

The relationship between the average ROP and hot water flow rate for three different hot water temperatures is shown in Figure 11. It can be readily seen that, at each water temperature, as water flow rate increased, the drill penetrated at a faster rate. Additionally, the ROP increased when the water temperature was higher. For the case of 70 °C water temperature, the ROP reached 27.8 m/h.

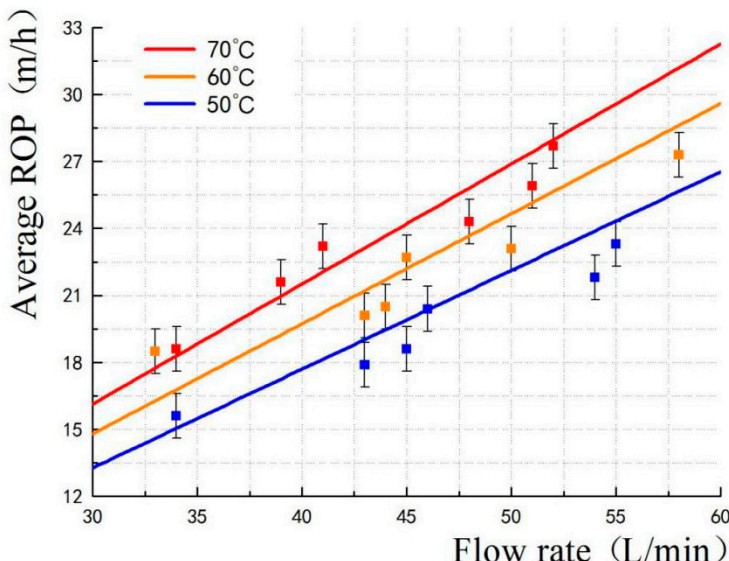

**Figure 11.** The comparison between experimental and theoretical data. The lines were drawn using Equation (8).

During the calculation of ROP, the average diameters of borehole and ice core at the same temperature obtained in the test runs ($D_{50}$ = 237 mm, $d_{50}$ = 53 mm, $D_{60}$ = 257 mm, $d_{60}$ = 53 mm, $D_{70}$ = 273 mm, $d_{70}$ = 45 mm) were used as the values for D and d in Equation (8). Thus, according to Equation (8) and all the values from our test results, the theoretical relationships between ROP and hot water flow rate at 50/60/70 °C of hot water are also shown in Figure 11.

Table 2 shows the error analysis between experimental and theoretical results, and because the maximum RMSE is quite small, we consider that the experimental and theoretical results were in good agreement.

**Table 2.** RMSE analysis between experimental and theoretical results.

| Temperature (°C) | Root Mean Squared Error |
|---|---|
| 70 | 1.23 |
| 60 | 1.20 |
| 50 | 1.21 |

## 7. Conclusions

A hot water ice-coring system to be used in combination with the Chinese Hot Water Drilling System was designed, tested, and some calculations with it were performed.

(1) A hot water ice-coring drill was designed with a special mechanism that can redirect the flow of hot water upwards, that allows the borehole to be enlarged during back drilling to prevent borehole closure from happening.

(2) The drill was tested many times in the ice drill testing facility in Jilin University, China. Test results showed that the drill worked adequately to recover ice cores.

(3) Relations between ROP and hot water flow rates at different water temperatures were obtained from analysis of the test results. Choosing optimal values for the operation parameters is often a trade-off among different objectives, such as penetration rate and large size of ice cores. The trade-off temperature for the ice-coring drill was 50 °C, and the trade-off water flow rate was between 42 L/min to 55 L/min, which allowed acquisition of ice cores of the best quality (the maximum diameter of ice core).

(4)    Theoretical estimations were also performed and compared with the test data. Comparison showed that there was a good agreement between theoretical calculation and experimental results, which indicates that the theoretical estimation can be a good reference when preliminary assessment of the Chinese Hot Water Drilling System is performed in the field.

**Author Contributions:** The statements of author contributions are as following: conceptualization: P.T., R.W.; methodology: X.F. and P.T.; formal analysis: R.W., A.L; writing—original draft: A.L.; writing—review and editing: P.T., X.F., Y.Y.; validation, P.T., R.W., X.F. and Y.Y.; investigation, A.L., X.L. and L.W.; project administration, P.T.

**Funding:** This research was supported by the National Science Foundation of China (Projects No. 41476160 and No. 41327804) and by the Science and Technology Innovative Research Team of Jilin University (Project No. 2017TD-24). The authors would like to acknowledge the contribution of the Polar Research Center of China for science investigation on hot water ice-drilling system. The authors also would like to show great gratitude to Prof Engelhardt for his innovative work in hot water ice-coring drill design.

**Conflicts of Interest:** The authors declare no conflicts of interest.

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
