# Peer review of "Test-Bed Performance of an Ice-Coring Drill Used with a Hot Water Drilling System"

_jmse, doi:10.3390/jmse7070234_

Reviewer 1 Report

The language appears to have improved significantly. There are still minor issues concerning grammar and syntax.

In table 1, should the unit for 'Length of core' be 'mm'?

I have no further comments.

Author Response

Point 1: In table 1, should the unit for 'Length of core' be 'mm'?

Response 1: I have revised the mistake about unit and changed the the unit of 'mm' to the unit of 'm'.

Reviewer 2 Report

This manuscript describes a modern development of an ice coring drill that could be used in a hot-water drilled borehole in ice sheets.  The idea isn’t new – hot-water drilling of access boreholes particularly in Antarctica has been going on for several decades, and is the most rapid drilling method available.  Although almost routine for access drilling,  rarely has anyone tried to collect cores from the ice sheet at selected depths.  Generally, ice cores are nowadays recovered with electromechanical drills designed to cut an annulus in the ice and recover the chips and core together in either dry shallow boreholes, or low-temperature fluid filled boreholes for greater depth.  The strength of ice core research normally lies in recovering continuous cores from the surface to great depth in order to generate a full depth profile for research into climate and atmospheric history.  Since discrete short sections of ice from various depths are very difficult to date independently of the continuous evolving climate /atmosphere history from the surface, ice core science has not placed a great deal of emphasis on recovering discrete sections of ice as described in this paper.  

That said, there are occasions where the main purpose of a drilling project is to provide a rapidly drilled deep access hole (either to the bed of the ice sheet or to a sub-glacial lake, or through an ice shelf for access to the ocean below), and where there is significant interest in  recovering short sections of the ice for quite specific purposes.  And this is where the technology described in this paper provides a solution.  

The paper does describe the engineering of the hot-water ice drill well, and through testing in a non-polar facility shows that the actual performance of the drill largely matches a priori calculations of likely performance in terms of penetration rate.

The paper does need a lot of polishing on the English.  I could understand well enough the descriptions, but in many places this was because I am familiar with the technology described. I’m less sure the general reader would find it as easy.  

Generally, the paper is logical, and detailed enough to understand the motivation, design and performance characteristics of the drill. I’ve several minor comments detailed below.

Title: I think the title should reflect that the testing took place only in a test-bed facility, and not in the field.  ‘Test-bed performance of an ice-coring…..’ perhaps?

Line 24 – ice cores through ‘ice shelves’ are actually quite rare. Cores are normally collected from ‘ice sheets’ for climate research.  (‘Ice sheet’ is ice resting on bedrock, whereas ‘ice shelf’ is floating ice that has flowed off ice sheets and into the ocean.)  I believe the authors are using ‘ice shelves here because the drill has been developed to take advantage of Chinese projects to drill the Amery Ice Shelf.

Line 28-33 - I think it is unreasonable to suggest that these drawbacks of traditional electromechanical ice core drilling are going to be solved by hot-water ice core drilling.  I can tell you that no mainstream ice core scientist is going to move from electrotechnical to hot-water drilled cores.  For example, to criticise traditional ice coring for ‘fragmented ice cores’ is a bit much when it is clear that the hot-water drilled cores will be recovered only very infrequently in the borehole depth.  This isn’t just because the main purpose of the drilling is likely to be for access, but for the very fact that the hot-water drilled borehole is continually cooling and freezing, and in the time taken to deploy the hot-water core drill, the likelihood is that the borehole will have cooled enough to need reaming.  It’s simply not possible to recover more than a few cores in hot-water drilling – the energy budget doesn’t allow it.  There is a purpose for hot-water core drilling, but it isn’t to replace electromechanical core drilling, whatever its drawbacks.

A lot is made in the paper of the speed of penetration, and I suppose this is made to imply the method is fast compared to electromechanical drilling.  While it is the case that the coring rate is faster, these rates are simply snapshot rates when the core drill is working, and this is not how the drill penetrates the ice depth – the majority of penetration of the ice depth comes during the normal hot-water access hole drilling.  Penetration rate of ice core drilling is a combination of cutting speed, but also winching speed to recover the core each run.  So, give the penetration rate, but don’t suggest this makes it a rapid method of collecting cores in general.

Line 43 – this is pointing to US drilling projects, not British. The reference in line 45 is to a US project, and the UK had not drilled to more than 2000m until 2019.

Line 145 – it is not clear why the final water temperature is +20C. Surely the water continues to cool as it loses energy to melt more ice, and approaches 0 degrees.  This needs some further explanation – presumably you are pumping the water away from the head, but how do you control that it is pumped away at 20C?

Line 199, table 1.  Length of core is surely in m not mm.  I would not sort the table on run number (which is of no interest), but sort first on water temperature, and secondly on flow rate.  This give an easier way into looking at the data in detail, and comparing line to line.

Line 206 and 225, figure 10 and 12.  Do we need both?  The penetration rate observed is the same in both (and also in table 1).  I think only figure 12 is needed.
